# High-Resolution Taxonomic Characterization Reveals Novel Human Microbial Strains with Potential as Risk Factors and Probiotics for Prediabetes and Type 2 Diabetes

**DOI:** 10.3390/microorganisms11030758

**Published:** 2023-03-15

**Authors:** Sarah A. Hendricks, Chantal A. Vella, Daniel D. New, Afiya Aunjum, Maximilian Antush, Rayme Geidl, Kimberly R. Andrews, Onesmo B. Balemba

**Affiliations:** 1Institute for Interdisciplinary Data Sciences, University of Idaho, Moscow, ID 83843, USA; 2Department of Movement Sciences, University of Idaho, Moscow, ID 83843, USA; 3WWAMI Medical Education Program, University of Idaho, Moscow, ID 83843, USA; 4Department of Biological Sciences, University of Idaho, Moscow, ID 83843, USA

**Keywords:** fecal microbiota, gut microbiota, hyperglycemia, insulin resistance, long-read sequencing, prediabetes, type 2 diabetes

## Abstract

Alterations in the composition of the gut microbiota is thought to play a key role in causing type 2 diabetes, yet is not fully understood, especially at the strain level. Here, we used long-read DNA sequencing technology of 16S-ITS-23S rRNA genes for high-resolution characterization of gut microbiota in the development of type 2 diabetes. Gut microbiota composition was characterized from fecal DNA from 47 participants divided into 4 cohorts based on glycemic control: normal glycemic control (healthy; *n* = 21), reversed prediabetes (prediabetes/healthy; *n* = 8), prediabetes (*n* = 8), or type 2 diabetes (*n* = 10). A total of 46 taxa were found to be possibly related to progression from healthy state to type 2 diabetes. *Bacteroides coprophilus* DSM 18228, *Bifidobacterium pseudocatenulatum* DSM 20438, and *Bifidobacterium adolescentis* ATCC 15703 could confer resistance to glucose intolerance. On the other hand, *Odoribacter laneus* YIT 12061 may be pathogenic as it was found to be more abundant in type 2 diabetes participants than other cohorts. This research increases our understanding of the structural modulation of gut microbiota in the pathogenesis of type 2 diabetes and highlights gut microbiota strains, with the potential for targeted opportunistic pathogen control or consideration for probiotic prophylaxis and treatment.

## 1. Introduction

Gut bacterial populations are prone to perturbations driven by intrinsic host factors (e.g., genetics and lifecycle stage) and environmental factors (e.g., pollutants, diet, lifestyle, pharmaceutical use) that can alter host physiology and health status [1]. Alterations in the gut microbiota composition (dysbiosis) have been associated with the progression of insulin resistance (IR) and type 2 diabetes [2]. Among consistently reported findings, the signatures of dysbiosis associated with type 2 diabetes include, but are not limited to, a decrease in the genera of *Bifidobacterium*, *Bacteroides*, *Faecalibacterium*, *Akkermansia*, and *Roseburia*; and an increase in the genera of *Ruminococcus*, *Fusobacterium*, and *Blautia* [3]. Several studies have found a correlation between type 2 diabetes and gut microbiota alpha (α)-diversity, as well as *Firmicutes/Bacteroidetes* (F/B) ratios; however, other studies have found no correlation [3]. Other commonly found gut microbiome shifts include the depletion of butyrate-producing bacteria [4] and a reduction in probiotics [5]. For example, several bacterial species, such as *Lactobacillus fermentum*, *L. plantarum* and *L. casei*, *Roseburia intestinalis*, *Akkermansia muciniphila*, and *Bacteroides fragilis*, have been shown to decrease the risk of diabetes development through maintaining intestinal barrier integrity, improving glucose metabolism and insulin sensitivity, and suppressing proinflammatory cytokines [6]. While these protective species may be candidates for probiotics, it is essential that they be carefully characterized and assigned to a given strain, and not to the entire genus or species [7].

To date, strain level data have been limited in studies identifying bacterial changes accompanying metabolic disease. Fei and Zhao (2013) found that a single strain, *Enterobacter cloacae* B29, in combination with a high-fat diet, induced fully developed obesity phenotypes, including endotoxin-induced inflammation, adiposity, and IR in gnotobiotic mice [8]. Aside from this case, most strain level microbiome research related to metabolic disease is focused on probiotics. For example, a comparison between *Bifidobacterium animalis* ssp. *lactis* GCL2505 (BlaG) and *B. longum* ssp. *longum* JCM1217T found that BlaG reduced visceral fat accumulation and improved glucose tolerance in a mouse model [9]. Without high resolution taxonomic information, these studies would not have been able to identify strains for development of effective probiotics. Further strain-level information regarding type 2 diabetes progression could be useful for targeted opportunistic pathogen control or probiotic prophylaxis and treatment.

In this study, we aimed to identify novel strains associated with type 2 diabetes. We investigated the fecal microbiota in humans with prediabetes and type 2 diabetes, as well as those without diabetes. Using long-read DNA sequencing technology, a high-resolution ~2500-bp genetic marker (including the entire 16S gene and two additional genes), and a recently published reference database [10], we estimated microbial abundance and diversity, assessed microbiota associated with health status, and report new stains associated with type 2 diabetes progression.

## 2. Methods

### 2.1. Consent and IRB

A convenience sample of 48 adults (mean ± SD age 51.0 ± 15.9 years, 60% women) with and without type 2 diabetes, aged 18 years and older, were recruited from a university and the surrounding community to participate in this study that included two visits to the laboratory for each patient. Eligibility for the study was determined by a pre-screening questionnaire that was completed over the phone by a trained researcher. Study exclusion criteria included a history of gastric bypass surgery, inflammatory bowel disease (i.e., irritable bowel syndrome, Crohn’s or colitis), colon cancer, celiac disease, multiple sclerosis, Parkinson’s disease, Alzheimer’s disease, or current pregnancy. The University of Idaho Institutional Review Board approved the study (protocol 20-098) and all participants provided verbal and written informed consent to participate in the study. All experiments were conducted in accordance with the Declaration of Helsinki.

### 2.2. HbA1c Measurement

A small sample of blood (~1 µL) was obtained from the middle or ring finger of the non-dominant hand and analyzed for HbA1c under strict standardized operating procedures using the DCA Vantage analyzer (Siemens Healthcare Diagnostics, Tarrytown, NY, USA). In brief, the finger was cleaned with alcohol prior to measurement. A disposable safety lancet (SurgiLance, MediPurpose, Duluth, GA, USA) was used to puncture the skin and the first drop of blood was wiped clean. The sample was drawn into the capillary tube, inserted into the reagent cartridge, and analyzed immediately. Participants were divided into four groups, based on measured hemoglobin A1c (HbA1c) during the first study visit: healthy (*n* = 21, HbA1c < 5.7%), healthy range but previously diagnosed with prediabetes by a registered physician (prediabetes/healthy (reversed prediabetes), *n* = 9; HbA1c < 5.7%), prediabetes (*n* = 8, HbA1c 5.7–6.4%), and type 2 diabetes (*n* = 10, HbA1c ≥ 6.5%).

### 2.3. Stool Collection

After collection of the blood sample, participants were provided a stool sample kit and detailed instructions to collect and store the sample. All samples were stored on ice in a Styrofoam cooler, brought to the laboratory within 24 h of collection, and frozen at −80 °C prior to extraction. The average number of days between the blood sample collection and fecal sample collection was 4.8 ± 2.1 days (range 1–8 days).

### 2.4. DNA Isolation, Sequencing, and Taxonomic Identification

Total genomic DNA from 250 mg of human fecal samples was extracted using the QIAamp PowerFecal Pro DNA kits (Qiagen, Hilden, Germany) according to the manufacturer’s instructions. The DNA concentration and purity were monitored on 1.5% agarose gels. The Intus Biosciences Wave StrainID Kits (Farmington, CT, USA), SetA and SetZ, were then used to produce amplicons that span the full-length 16S, ITS, and partial 23S rRNA genes. Sequencing libraries were generated from these amplicons using PacBio SMRTbell Express Template Prep Kit v.3.0 (Pacific Biosciences, Menlo Park, CA 94025, USA) according to the manufacturer’s instructions. We included two negative controls (elution buffer and DNA extraction with water). The library was sequenced on 1 SMRT Cell 8M on a PacBio Sequel II System using the circular consensus sequencing (CCS) mode at the University of Idaho Institute for Interdisciplinary Data Sciences Genomics and Bioinformatics Resources Core. The CCS reads were determined with a minimum predicted accuracy of 0.9 and the minimum number of passes set to three in the SMRT Link software package v.10.2.1.143962 (Pacific Biosciences).

SBanalyzer v.3.0 (Intus Biosciences) was used to demultiplex and assign taxonomic identification to all reads by mapping to the Athena database [10]. In this analysis, sequences are clustered by similarity with a threshold of 97% and put into bins called ‘Operational Taxonomic Units’ (OTUs). Setting the sequence similarity threshold to 97% to delineate species is only a rough approximation and, in some cases, may not be biologically accurate to distinguish between some taxa [11]. A single sequence is selected as a representative sequence for each of the OTU bins. If a read does not match any strains or if it matches equally to multiple strains in the database, the taxonomic classification will be reported at the highest level where an unambiguous call can be made, resulting in an “unclassified” taxon. The output reports abundance for each OTU for each patient. Patients with less than 500 reads were removed from all subsequent analyses. Potential reagent contaminants were identified using the decontam package v.1.14.0 with a threshold frequency of 0.5 of the OTU in the negative control [12,13]. The decontam package has been shown to remove upward of 90% of contaminants even when the source of contamination was not well defined [14]. To determine whether sequencing depth was sufficient to capture a nearly complete microbiome profile for samples, rarefaction curves were plotted using the R-package vegan (v.2.6.2; [15]).

### 2.5. Alpha and Beta Diversity Analyses

Subsequent analysis of the filtered OTU table was conducted in R [16] using the phyloseq package v.1.16.2 [17]. To calculate and plot α-diversity indices (Shannon diversity, Simpson index, and Chao1 index) of the microbiome communities, we used phyloseq. To analyze whether the α-diversity was significantly different between the four health status groups (healthy, prediabetes/healthy, prediabetes, diabetes), we first tested for normal distribution of each of the α-diversity indices using the Shapiro–Wilk test of normality. For nonparametric data, we tested for significant differences using the Kruskal–Wallis test. For normally distributed data, we used an ANOVA test and subsequent Tukey’s honest significance test of the ANOVA to test for statistical differences.

We calculated beta (β) diversity using non-metric multidimensional scaling (NMDS) from Bray-Curtis dissimilarity in phyloseq. To test whether microbial communities differ by donor health status, we used the adonis2 function in R-package vegan (v 2.6.2) to run a permutational multivariate analysis of variance (PERMANOVA) after testing for multivariate homogeneity of group dispersion (beta dispersion).

### 2.6. Identification of Differentially Abundant Taxonomic Groups

Community composition barplots were generated using the fantaxtic v.0.2.0 package in R. To identify taxa that were significantly different between health status groups, we used the DESeq2 package as described for microbiome applications [18,19]. After controlling for donor sex in the design matrix, DESeq2 was run under the Wald significance tests and parametric fitting of dispersions to the mean intensity settings, and false discovery rate adjusted p-values were calculated with the Benjamini-Hochberg procedure using a significance threshold of *p* < 0.05. Differentially abundant (or “discriminatory”) taxa were assessed at the phylum, genus, and strain levels. To further investigate the level of differentiation across groups for discriminatory strains, we generated NMDS plots using Bray–Curtis dissimilarity for strain-level OTUs that were differentially abundant for one or more pairwise comparisons.

## 3. Results

### 3.1. DNA Isolation, Sequencing, and Taxonomic Identification

After removing one individual due to a low number of sequencing reads (SG11), the number of fecal microbial DNA samples sequenced was 21 healthy, 8 prediabetes/healthy, 8 prediabetes, and 10 type 2 diabetes individuals (Appendix A). The mean number of CCS reads per sample was 15,836.76. After all filtering steps, 641,305 reads were successfully classified with a mean of 13,644.79 (±5311.93) per sample.

Before removing contaminants, 884 OTUs were identified. Eight OTUs were removed at a prevalence threshold of 0.5. After removing contaminants, ten phyla were identified, including *Actinobacteria*, *Bacteroidetes*, *Cyanobacteria*, *Elusimicrobia*, *Firmicutes*, *Fusobacteria*, *Proteobacteria*, *Synergistetes*, *Tenericutes* (*Mycoplasmatota*), and *Verrucomicrobia*. Four phyla, namely *Firmicutes*, *Bacteroidetes*, *Actinobacteria*, and *Proteobacteria* were the most abundant across all samples (Figure 1), with a mean percent total reads across samples for these phyla of 58.80% (±16.85), 23.13% (±15.90), 10.72% (±10.37), and 5.15% (±5.76), respectively (Appendix A). These phyla were represented by 29 classes, 62 orders, 114 families, 312 genera, 602 species, and 876 strains. Of the 876 strains, 314 were known, previously published strains and 562 were unclassified (de novo) strains (Appendix A).

### 3.2. Alpha and Beta Diversity

Rarefaction curves of the gut microbiomes of all individuals approached, but did not reach, a plateau, indicating that our dataset likely captured the dominant patterns of the microbial communities; however, that increased sequencing depth may have captured additional information regarding low-frequency taxa (Appendix A). We used the Chao1 index as an indicator of microbial community richness, and Shannon and Simpson indexes as indicators of diversity (Figure 2a). These alpha diversity indices were not significantly different between the bacterial gut microbiomes of individuals from any health status comparisons. The Shapiro–Wilk normality test indicated that the Shannon and Simpson diversity values were not normally distributed (Shannon: W = 0.92, *p* = 0.003; Simpson: W = 0.79, *p* = 1.13 × 10^−6^), and the Kruskal–Wallis test showed no evidence of differences between groups (Shannon: chi-squared = 2.65, df = 3, *p* = 0.45; Simpson: chi-squared = 2.02, df = 3, *p* = 0.57). Chao1 richness was normally distributed (W = 0.99, *p* = 0.91) and was not significantly different between groups (*p* = 0.31).

When testing for β-diversity, the NMDS plot did not indicate distinct clustering between health status cohorts, as evidenced by overlapping ellipses (Figure 2b). However, the PERMANOVA test did identify significant differences (*p* < 0.05) between health status categories (F = 1.72; R^2^ = 0.11; *p* = 0.02).

### 3.3. Differentially Abundant Taxa

Some taxa exhibited significant differences (*p* < 0.05) in abundance between health status cohorts. At the phylum level, two phyla were less abundant in the type 2 diabetes cohort than other cohorts (Appendix A). In particular, *Tenericutes* was less abundant in type 2 diabetes participants compared to all other cohorts, and *Firmicutes* was less abundant in type 2 diabetes participants compared to healthy and prediabetes/healthy participants. *Proteobacteria* were more abundant in healthy and prediabetes/healthy participants (Appendix A). No phyla exhibited abundance differences for the remaining cohort comparisons (prediabetes to healthy, prediabetes/healthy to healthy, and prediabetes/healthy to prediabetes). F/B ratios were significantly higher for type 2 diabetes than for any of the three other cohorts (healthy controls: *p* = 0.01; prediabetes/healthy: *p* < 0.001; prediabetes: *p* = 0.01; Appendix A). There were no significant differences in F/B ratios in any other comparisons (prediabetes to healthy: *p* = 0.35, prediabetes/healthy to healthy: *p* = 0.06, and prediabetes/healthy to prediabetes: *p* = 0.57).

At the genus level, several genera were significantly more abundant for the healthy cohort than for participants in type 2 diabetes (5 genera) and prediabetes (3 genera) cohorts (*p* < 0.05, Appendix A). In contrast, *Streptococcus* and (*Bacteroides*) genera were significantly less abundant in healthy individuals than in prediabetes/healthy and prediabetes participants, respectively. We also observed that one genus (*Lactococcus*) was less abundant and two (*Selenomonas and Megasphaera*) were more abundant in prediabetes/healthy participants compared to prediabetes participants. Prediabetes participants had five genera more abundant than in type 2 diabetes participants, while type 2 diabetes participants had four genera more abundant than prediabetes participants (Appendix A). There were three genera more abundant in type 2 diabetes as compared to prediabetes/healthy and four genera less abundant between the two cohorts (Appendix A).

At the OTU level, 55 distinct taxa were found to differ in abundance across cohorts, with 39 of those taxa found to be significantly different in more than one comparison, and 16 taxa unique to a single comparison. Of these 55 OTUs, 60% were identified to species level (*n* = 15) or strain level (*n* = 18). The number of differentially abundant OTUs ranged between 10 and 33 for the six pairwise comparisons of treatment cohorts (Figure 3, Appendix A). When calculating β-diversity using only strains that were differentially abundant across one or more pairwise comparisons, NMDS primarily distinguished healthy from diabetic participants (Appendix A).

## 4. Discussion

In the present study, the fecal microbiota of healthy, prediabetes/healthy, prediabetes, and type 2 diabetes individuals were explored using long-read technology to sequence 16S-ITS-23S amplicons, and taxonomic classification was performed using a recently published reference database. The gut microbiota structure of all individuals was found to be mainly composed of *Bacteroidetes*, *Firmicutes*, *Proteobacteria*, and *Actinobacteria*. To our knowledge, we identified more strain-level microbial differences associated with type 2 diabetes progression than previous studies were able to find due to methodical limitations resulting in a lack of species- and strain-level resolution. As with many previous studies, we noted bacteria that may act as opportunistic pathogens as they are more abundant in the diabetic group than the glucose tolerant groups. Further, our study revealed that some taxa are significantly correlated with healthy individuals and likely infer a protective or probiotic role within the host, which corroborates findings from other studies that show the importance of these microbial markers to health.

### 4.1. Probiotic Taxa

We identified several taxa that were positively associated with healthy status and that could be candidates for probiotic development. *Bifidobacterium pseudocatenulatum* (unclassified and strain DSM 20438) was higher in abundance with healthy status and, similarly, *B. adolescentis* ATCC 15703 shifts to a greater abundance in the healthy group as compared to prediabetes/healthy. There is strong support for *Bifidobacterium*’s protective role in type 2 diabetes in previous studies. Many papers report a negative association between this genus and type 2 diabetes (see [3]), with only one paper reporting conflicting results [20]. Certain species, such as *B. adolescentis*, *B. bifidum*, *B. pseudocatenulatum*, *B. longum*, and *B. dentium*, have been found to have a negative association with type 2 diabetes patients treated with the antihyperglycaemic agent metformin [21]. In animal studies testing for probiotic effects, several species/strains from this genus (*B. bifidum*, *B. longum*, *B. infantis*, *B. animalis*, *B. pseudocatenulatum* CECT 7765, and *B. breve*) showed improvement of glucose tolerance [9,22,23,24,25], with some species (*B. adolescentis*) being superior in alleviating type 2 diabetes symptoms compared to other species (*B. bifidum*; [26]). Our results support the notion that *Bifidobacterium* naturally inhabiting the human gut may play a protective role in type 2 diabetes and indicate that two strains, *B. pseudocatenulatum* DSM 20438 and *B. adolescentis* ATCC 15703, have the potential for new probiotic agents.

We also found *Prevotella copri* DSM 18205 to be more abundant as health increases. However, there have been contradictory results regarding the pathophysiological role of *P. copri.* This species is correlated with IR in humans and in fat-fed mice [27], and has been found to be elevated in type 2 diabetes [28]. Direct inoculation of rats with *P. copri* showed an improved glucose homeostasis [29]. This inconsistent pattern may be due to the existence of several *P. copri* clades or host diet-dependent effects [30]. Given that these microbes were more abundant in glucose tolerant groups, they may be possible candidates for probiotic use in type 2 diabetes prevention and treatment. However, risk, efficacy, and dosage for each strain should be fully explored through science-based clinical studies on targeted populations and pre-market approvals should be established [31].

We found that *Romboutsia*, a butyrate producing bacteria genus, and *R. timonensis* also increased in healthy and prediabetes individuals as compared to the type 2 diabetes group. Similarly, other studies found *Romboutsia* to be inversely associated with IR, type 2 diabetes, and gestational diabetes mellitus [32,33,34]. Additionally, studies have shown that *Romboutsia* is negatively correlated with fasting glucose, insulin, and high-density lipoprotein cholesterol; and positively correlated with indicators of obesity [35]. Contrary to these findings, one study found a significantly lower relative abundance of *R. timonensis* in type 2 diabetes participants as compared to healthy controls [36]. Further, Hu et al. (2019) reported that *Romboutsia* was reduced in the diabetic rats [37]. Therefore, *Romboutsia* may be a candidate genus for predicting and treating obesity and related metabolic disorders; however, more species and strain-specific knowledge is required to define utilities of these microbes in humans.

### 4.2. Pathogenic Taxa

In this study, we found several taxa that increased in abundance with progression from prediabetes to type 2 diabetes, indicating a possible pathogenic effect. *Odoribacter laneus* YIT 12061 was found to be significantly more abundant in prediabetes/healthy, prediabetes, and type 2 diabetes than in healthy controls. This species is a succinate-consuming bacterium and has been associated with a number of diseases, including colitis in a mouse model [38]. *Odoribacter* has been found to be negatively associated with the expression of intestinal epithelial tight junction proteins, suggesting a link with impairing mucosal barrier function in rats [39]. However, Huber-Ruano et al. (2022) showed that one strain of *O. laneus*, DSM 22474 A, has beneficial anti-inflammatory and metabolic properties in obese mice [40]. Collectively, our results strongly suggest that the effects of this species may be strain-specific and indicate the need of comprehensive evaluation before pursuing *Odoribacter* strains for treating type 2 diabetes in humans.

Another genus and species that increased in abundance and could be involved in the development of type 2 diabetes were *Flavonifractor plautii* and an unclassified *Flavonifractor* species, which were more abundant in type 2 diabetes participants than prediabetes/healthy participants. This pattern supports previous findings showing that the genus *Flavonifractor* decreased in prediabetes and increased in type 2 diabetes [41], and was associated with a lower insulin sensitivity and higher prevalence of dysglycemia [42]. *F. plautii*, specifically, was lower in patients with prediabetes and higher in patients with diabetes [41,43]. One possible mechanism underlying the association of *Flavonifractor* with type 2 diabetes progression is that this genus may increase oxidative stress and trigger inflammatory cytokines in plasma [44,45]. These cytokines are associated with type 2 diabetes by impairing insulin signal transduction and triggering IR [46]. To support this, possible treatments such as the oral administration of *Sanghuangporous vaninii* SVE have been shown to improve body weight, glycolipid metabolism, and inflammation-related parameters through modulating gut microbiota including decreasing *Flavonifractor* [47].

### 4.3. Species- and Strain-Dependent Associations

We identified stain-dependent type 2 diabetes associations within the *Escherichia* genus. *Escherichia coli* O128:H27 were more enriched in healthy and prediabetes/healthy than in prediabetes and type 2 diabetes. In contrast, *E. coli* O25b:H4 and *E. coli* O25b:H4-ST131 were more abundant in type 2 diabetes and prediabetes than in prediabetes/healthy; however, both were more abundant in healthy than prediabetes/healthy. Following a similar pattern, *E. coli* O1:H42 was more abundant in type 2 diabetes than prediabetes and healthy, but also more abundant in healthy as compared to prediabetes. We discovered an unclassified strain of *E. coli* was more abundant in type 2 diabetes than in prediabetes/healthy. Previous studies have positively associated *Escherichia* increasing in abundance with progression from normal glucose tolerance to type 2 diabetes [48,49,50] and with effects of metformin therapy [21,51]. Our results support the findings that *E. coli* may play a role in the pathogenesis of type 2 diabetes as well as metformin mechanisms of action. However, our work illuminates the essence of *E. coli* strain specificity when considering their functional use as antidiabetic targets for humans.

Within the genus *Bacteroides*, *B. coprophilus* DSM 18228 = JCM 13818, and *B. stercoris* ATCC 43183 were more abundant as health increases; however, *B. vulgatus* (unclassified strain and strain ATCC 8482) were higher in type 2 diabetes participants. Moreover, (*Bacteroides*) *pectinophilus* ATCC 43243 was unique to prediabetes and *B. coprocola* DSM 17136 to prediabetes/healthy individuals. According to Gurung et al. (2020), many previous studies have found *Bacteroides* to be either negatively or positively associated with type 2 diabetes [3]. Our study suggests that these inconsistencies could be driven by species and strain-specific effects, which would not have been distinguished in previous studies due to limitations of taxonomic classification below the genus level with traditional partial-16S sequencing. Alternatively, other confounding variables could be involved. For example, the studies that found positive associations with disease noted that participants had taken antidiabetic treatments, such as metformin, which have been shown to result in functional microbial shifts [51,52]. This could account for the apparent inconsistency in this study as well as previous studies. There could be several other confounding variables resulting in these inconsistencies, such as underlying host genetics, diet, physical activity level, medication use, and sequencing techniques. Nonetheless, our study indicates that species specificity of *Bacteroides* is important when considering the utility of antidiabetic properties and as a beneficial role on glucose metabolism in humans.

We identified three *Lactobacillus* species that were differentially abundant in our association analysis. *Lactobacillus animalis* and *L. murinus* are more abundant in both the two extreme phenotypes, healthy and type 2 diabetes, as compared to the prediabetes/healthy and prediabetes participants, whereas *L. johnsonii* was associated with type 2 diabetes. Although *L. animalis* and *L. murinus* were found to be highly abundant in healthy individuals, they both were also highly abundant in type 2 diabetes, so are not likely candidates for probiotics. The highly diverse *Lactobacillus* genus, which is frequently detected and reported in type 2 diabetes association studies, has the highest number of taxa with probiotic potential found in the human gut to date [3]. The effects of this genus on type 2 diabetes seem to be species-specific. For example, three species, *L. acidophilus* [53], *L. gasseri* [54], *L. salivarius* [54], were significantly more abundant in type 2 diabetes participants, while *L. amylovorus* [51] was decreased in type 2 diabetes participants. These previous results, as well as the results of our study suggests that species and strain specificity from this genus impact the functionality of host metabolism. Collectively, these differences in association patterns across species and strains reported herein highlight the significance of using high-resolution taxonomic methods to identify specific species and strain linking the microbiome with type 2 diabetes and other illnesses.

### 4.4. Microbial Diversity

Our study found mean richness and diversity measures were lower in participants with type 2 diabetes than all other participants, but higher in prediabetes/healthy and prediabetes than in healthy individuals. However, these differences were non-significant across comparisons, which may be due to limited sample size. Although α-diversity has been proposed as a biomarker in general health and disease, there is currently controversy regarding the association between diversity and type 2 diabetes due to inconsistent findings. For example, a systematic review of 13 studies investigating associations of gut microbiota α-diversity and type 2 diabetes found that an equal number of studies identified a negative correlation (*n* = 2) and a positive correlation (*n* = 2), and 9 found no correlation [3]. The lack of significant association for α-diversity with type 2 diabetes in our study supports the idea that gut microbiota richness and diversity may not necessarily decline with the progression of illness. However, many of the study participants classified as prediabetes/healthy and prediabetes were taking antihyperglycaemic agents, such as metformin. These agents have been found to increase the α-diversity of gut microbiota, as well as the reduction of HbA1c and fasting blood glucose concentrations [21,55]. These effects could have increased diversity in prediabetes/healthy and prediabetes participants relative to healthy participants in our study. It has also been noted that colonic transit time [56] and stool consistency [57] may interfere with the ability to use α-diversity as a biomarker in disease status.

Beta diversity, a measure of the similarity or dissimilarity of two communities, was found to statistically differ between healthy, prediabetes/healthy, prediabetes, or type 2 diabetes. According to Gurung et al. (2020), of 8 studies reporting β-diversity, 7 did not find any significant association between microbial β-diversity and type 2 diabetes [3]. Taxa may not have clustered distinctly on a community level as shown in the NMDS plot (Figure 2b) due to host participants living in relatively homogeneous environments [58] including similar host subsistence strategies, geographic location, and ethnicity [59].

### 4.5. F/B Ratios

The progression of type 2 diabetes may, in part, be due to the transformation of dominant bacteria, and, thus, F/B ratios are often measured to investigate associations between the microbiome and type 2 diabetes. The systematic review by Gurung et al. (2020) found that the B/F ratio did not show consistent associations with type 2 diabetes with six publications that found no association, three with a positive association, and four with a negative association [3]. Our findings showed that F/B ratios are highest in the type 2 diabetes group and significantly different than each of the three other categories. F/B ratios were not significantly different between healthy, prediabetes/healthy, and prediabetes cohorts likely due to the effect of antihyperglycaemic drugs as highlighted above.

### 4.6. Limitations and Future Directions

This study had several potential limitations. First, sample sizes were relatively small. Second, not all OTUs were resolved to the strain level, indicating that additional strain-level associations may exist in our dataset that could not be detected in our analyses. Thus, further development of the reference database could lead to even greater taxonomic resolution in future studies. Third, most prediabetes and type 2 diabetes participants in this study had taken antihyperglycaemic or anti-hypertensive drugs, and/or probiotics/prebiotics, which may have influenced the bacterial communities. Fourth, we could not control for every possible lifestyle factor or duration of disease in the study design, which leaves the possibility of residual confounding factors. Finally, our cross-sectional study cannot draw a direct link between bacterial taxa and particular metabolites involved in type 2 diabetes. Functional studies will be required to determine whether relevant bioactive metabolites were produced by the bacteria that were determined to be differentially abundant between diabetes participants and healthy participants, and if these metabolites were indeed causal for the disease phenotype. Future studies that integrate metagenomic and metabolomic approaches would greatly contribute to treatment discovery. Furthermore, a complete genome sequence and annotation, especially of genes associated with causal metabolites, is necessary for the registration of new probiotic strains.

## 5. Conclusions

To our knowledge, we identified more strain-level microbial differences associated with type 2 diabetes progression than previous studies were able to find due to methodical limitations resulting in a lack of species- and strain-level resolution. In addition, we demonstrated that sequencing a longer target region than the traditional partial-16S marker can improve taxonomic resolution and identify species-specific and strain-specific effects that could not otherwise have been discovered. The results of this work enhance our understanding of the influence of changes in human fecal microbiota on the pathogenesis of diabetes, and contribute insights into potential new microbiome biomarkers for preventative and/or therapeutic strategies for type 2 diabetes.

## Figures and Tables

**Figure 1 microorganisms-11-00758-f001:**
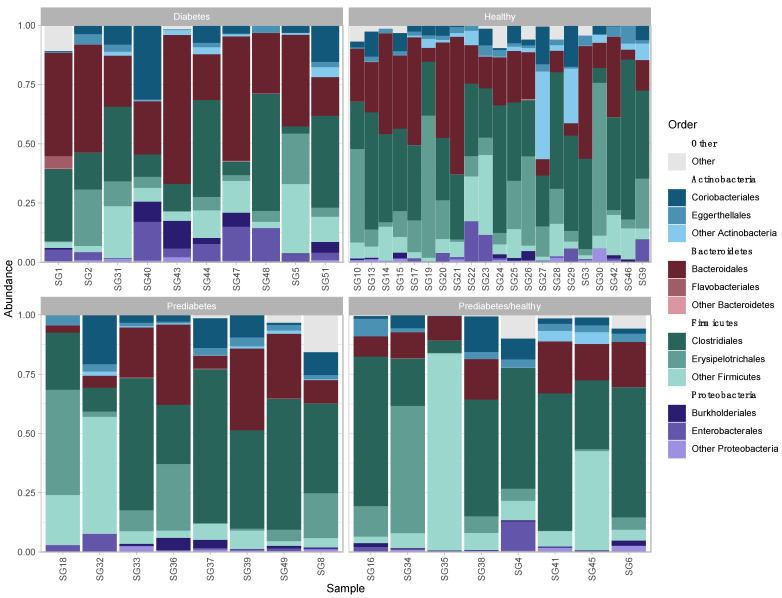
Relative abundance of the three most abundant bacterial orders within each of the four most abundant phyla in fecal microbiota of patients by health status.

**Figure 2 microorganisms-11-00758-f002:**
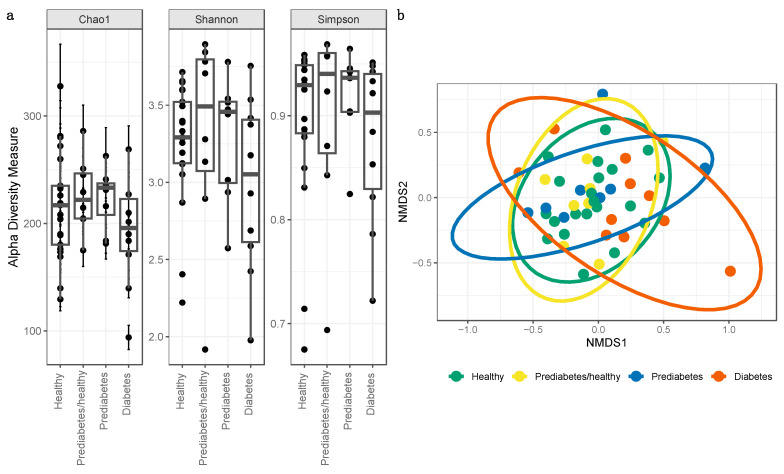
(**a**) Microbial alpha diversity as measured by the Chao1, Shannon, and Simpson index of fecal samples, which are colored according to four health status categories. All alpha diversity metrics were not statistically different between health status categories. (**b**) Microbial beta diversity. Dimensional reduction of the Bray-Curtis distance between microbiome samples, using NMDS ordination method, for fecal samples in healthy and controls. Data points are colored according to healthy status.

**Figure 3 microorganisms-11-00758-f003:**
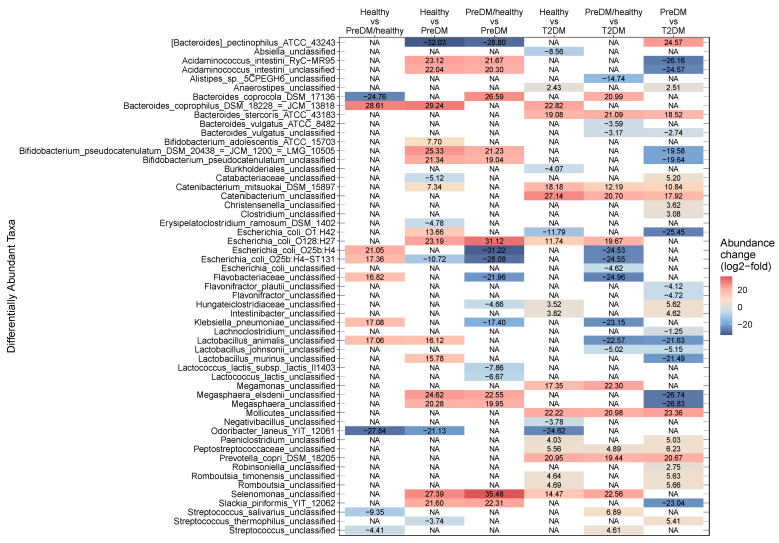
Observed fold change magnitudes for all 55 differentially abundant taxa present in all comparisons of health status groups. Columns indicate each comparison, and rows show taxonomic assignment for each OTU. Color indicates the log2-fold abundance change in each taxon for each comparison. For positive log2-fold change, the health status category before “vs” in each comparison label is more abundant than the category after the “vs”, and vice versa for negative log2-fold change. For example, when comparing “Healthy vs T2DM,” a positive value indicates a taxon that is more abundant in Healthy, and a negative value indicates a taxon that is more abundant in T2DM. T2DM: Type 2 diabetes; PreDM: prediabetes.

## Data Availability

The data presented in this study are openly available under BioProject accession number PRJNA930056. The SRA accession number for the reads are SAMN32982567 to SAMN32982614.

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
