# Peer review of "High-Resolution Taxonomic Characterization Reveals Novel Human Microbial Strains with Potential as Risk Factors and Probiotics for Prediabetes and Type 2 Diabetes"

_microorganisms, 2023, doi:10.3390/microorganisms11030758_

Round 1

Reviewer 1 Report

Dear editors:

The paper presented by Hendricks et al. is a very interesting document on the use of DNA sequencing technology of 16S-ITS-23S rRNA genes for high-resolution characterization of gut microbiota. The methodology used is highly new and could be useful for the detection and recognition of bacterial strains that could be applied as probiotics to control different diseases; in this work, Type 2 Diabetes is the target disease.

The document reflects a good study design, with only the negative point of a reduced number of samples, a condition also remarked by the authors. As I have mentioned previously, the methodology used is very interesting and highly satisfactory to obtain the expected results. Additionally, the methodology is accurately described, and could be useful to other research teams. The obtained results are very interesting, and the discussion correct.

Nevertheless, as I mention directly to authors, some additional points could be integrated in the discussion paragraph, in my opinion to enrich this section, before the approval of the document to its inclusion in the journal Microorganisms.

Author Response

Reviewer #1

Comment: I believe that the document that you present for its publication in the journal Microorganisms is in accordance with the aims of the journal and presents the use of the sequencing technology of 16S-ITS-23S rRNA genes for high-resolution characterization of gut microbiota, with the idea of arriving to the strain level in the microbiota analysis.

In my opinion, the article has an excellent quality, but I would introduce some additional points that could enrich your document, especially in the discussion paragraph.

The points that I propose that you take into consideration are related with the use of OTUs in the analysis of DNA sequences. I know that this concept is massively used in the studies on gut microbiota, but I believe that some aspects must be considered to improve the resolution of microbiota evaluation:

  1. Sequence analysis begins by clustering reads by sequence similarity into OTUs, an operational concept but in some cases not biologically supported. OTUs are generated by combining sequences that are similar at a defined percentage level (traditionally 97%). The level used in your presented work.

Our response: We thank the reviewer for underscoring the significance and strength of our manuscript. Also, we thank the reviewer for excellent comments. Per the published recommendations and the proprietary software for the type of data produced (StrainID), we used the 97% similarity level. We agree that in some cases this will not be biologically supported, so we added the statement: “Setting the sequence similarity threshold to 97% to delineate species is only a rough approximation and, in some cases, may not be biologically accurate to distinguish between some taxa[11].” Please, see, page 5, lines 147-149. The reference # 11 is “Nguyen N-P, Warnow T, Pop M, White B (2016) A perspective on 16S rRNA operational taxonomic unit clustering using sequence similarity. Npj Biofilms Microbiomes 2(1):1–8. https://doi.org/10.1038/npjbiofilms.2016.4”

  1. Comment: Some authors recommend the use of similarity level of 99% (Tikhonov et al., 2015) to achieve a sub-OTU resolution in deep sequencing. With this method it is possible to distinguish 20 subpopulations into each OTU obtained by the use of 97% similarity.

Our response: Due to the increased length of the sequence (~2,300bp) and the inclusion of the ITS region (which is more variable than 16S), as well as the high-resolution reference database, the protocol we used is able to resolve to a strain level at the 97% similarity. Increasing the similarity level to 99% may increase the likelihood of identifying false positives where sequencing errors are falsely being called as amplicon variants and therefore incorrectly identifying more subpopulations of each OTU. While this suggestion merits further exploration in a methodological experiment, we feel that it is out of the scope of the current manuscript.

  1. Comment: To study the DNA sequences of samples by the clustering-free analysis it is possible to use the files with the “denovo” sequences (the units used for the building of OTUs) created during the first steps of all the deep sequencing analysis methods. These files include the sequence, the taxonomic name at species level or at strain level, and the number of sequences with the same nucleotides chain.

Our response: The method we used to produce the data for the current manuscript uses a proprietary software and reference database due to the long-read length of the sequences. Any subsequent analysis on the de novo sequences would require running the raw sequence reads for each unclassified OTU through dada2 separately as recommended by Graf et al. (2021). Subsequent BLAST and phylogenetic analysis would provide information regarding possible closely related taxa and similarity. This is not feasible for 37 differentially abundant taxa. However, we have made the raw sequence data available on NCBI (Bioproject PRJNA930056). We also added an additional supplementary document that indicates the sample ID and taxonomic identification for each raw sequence read (Table S7).

Graf J, Ledala N, Caimano MJ, Jackson E, Gratalo D, Fasulo D, Driscoll MD, Coleman S, Matson AP. 2021. High-resolution differentiation of enteric bacteria in premature infant fecal microbiomes using a novel rRNA amplicon. mBio 12:e03656-20. https://doi.org/ 10.1128/mBio.03656-20.

  1. Comment: Differences between bacterial species and bacterial strain mentioned in “Differentially abundant taxa” paragraph, could be considered as artificial because are directly dependant on the OTU’s information incorporated in the database used for the analysis of DNA sequences. In some cases (Acidaminococcus intestini, Bifidobacterium pseudocatenulatum or Lactococcus lactis) the observed fold change magnitudes in all comparisons of health status groups, are similar between “unclassified” and “strain” taxa. In this examples, unclassified taxa could represent unknown strains highly similar to unclassified OTUs deposited in database.

Our response: We agree that some strains that are differentially abundant may not be reported in our study due to lack of completeness of the reference database, which would result in sequences being called “unclassified” below the species level. However, we do not expect that “classified” strains (i.e., strains identified to the strain level by SBAnalyzer) would artificially be identified as differentially abundant in our analyses. We now expand one of the sentences in our “Limitations” section to clarify that there may be additional strains with significant associations that could not be detected in our study. That sentence now reads: “Second, not all OTUs were resolved to the strain level, indicating that additional strain-level associations may exist in our dataset that could not be detected in our analyses. Thus, further development of the reference database could lead to even greater taxonomic resolution in future studies.” (Please, see, page 11, lines 438-442).

Comment: The relationship between F/B ratio and Type 2 Diabetes is a relevant correlation, as supported by the results obtained for other authors (Ley et al., 2006; Magne et al., 2020). The F/B ratio is a factor closely associated with intestinal dysbiosis in obese patients. As mentioned by Krajmalnik-Brown et al. (2012), Firmicutes are more effective in extracting energy from food than Bacteroidetes, conducting to a more efficient absorption of calories and the subsequent weight gain.

Our response: We agree that the relationship between F/B ratio and Type 2 Diabetes is a relevant correlation. We have revised the following paragraph in the discussion to be more specific as to what previous publications have found with regards to F/B ratio and Type 2 Diabetes: “The systematic review by Gurung et al. (2020) found that the B/F ratio did not show consistent associations with type 2 diabetes with 6 publications that found no association, 3 with a positive association and 4 with a negative association [3].” Please, see page 11, lines 429-431.

This statement includes some of the references the reviewer has stated. We chose to not include publications that looked for correlations between F/B ratios and obesity as we are focused specifically on Type 2 diabetes.

Comment: As you mention at the end of document, the inclusion of metabolomic studies could be very informative, because probiotic strains are closely associated with the expression of some metabolic pathways. Furthermore, for the registration of new probiotic strains a complete DNA sequencing and the complete annotation is necessary, especially the annotation of genes associated with antimicrobial resistances, and genes associated with metabolites that have positive effects on the health status.

Our response: Again, we thank the reviewer for this and other comments. We have included the additional point the reviewer has made in the “Limitations and future directions” paragraph. “Furthermore, a complete genome sequence and annotation, especially of genes associated with causal metabolites, is necessary for the registration of new probiotic strains.” Please, see page 11, lines 452-454.

Minor comments

Comment: Lines 8-10: The bacterial species must be in italic type letter.

Our response: All bacterial species have been corrected to be in italic type letter. According to American Society for Microbiology, strain designations and numbers are not printed in italics, which was the format we chose to use throughout the manuscript.

Comment: Line 175: In general, the breakpoint to consider significant a statistical analysis is 0.05. Why do you reduce this limit to 0.01 in the PERMANOVA test used to examine the effect of health status in the beta-diversity analysis?

Our response: We have corrected the p-value and any associated results and discussion as follows:

Results: “However, the PERMANOVA test did identify significant differences (p < 0.05) between health status categories (F = 1.72; R2 = 0.11; p = 0.02).” Please, see page 6, lines 225-226.

Discussion: “Beta diversity, a measure of the similarity or dissimilarity of two communities, was found to statistically differ between healthy, prediabetes/healthy, prediabetes, or type 2 diabetes. According to Gurung et al. (2020), of 8 studies reporting β-diversity, 7 did not find any significant association between microbial β-diversity and type 2 diabetes [3]. Taxa may not have clustered distinctly on a community level as shown in the NMDS plot (Fig. 2b) due to host participants living in relatively homogeneous environments [57] including similar host subsistence strategies, geographic location, and ethnicity [58].”

Comment: Line 266: Please, insert “possible” before “pathogenic effect”.

Our response: We have incorporated the recommended correction. Please, see page 8, line 322.

Reviewer 2 Report

This article is very interesting for the public of this journal. This study investigated the faecal microbiota in humans with prediabetes and type 2 diabetes, as well as those without diabetes, with the aim to identify novel strains associated with type 2 diabetes, highlighting that further strain level information regarding type 2 diabetes progression could be useful for targeted opportunistic pathogen control or probiotic prophylaxis and treatment. This study is very interesting and well designed and performed. A good work of analysts and methods was applied, and I suggest the publication.

- Line 30: ..Lactobacillus fermentum, plantarum and caseicorrect in L. plantarum and L. casei.

- Line 76,77: …during the first study visit: healthy (n = 21; HbA1c < 5.7%), healthy but diagnosed with prediabetes (prediabetes/healthy (reversed prediabetes), n = 9; HbA1c < 5.7%)…you can improve this sentence because is not clear the difference between these two class; it has the same percentage of HbA1c.

- I suggest you make an image with the results found, that explain easily the difference found in different cohorts of study, within resalt the taxa significative differences.

Author Response

Reviewer #2

Comment: This article is very interesting for the public of this journal. This study investigated the faecal microbiota in humans with prediabetes and type 2 diabetes, as well as those without diabetes, with the aim to identify novel strains associated with type 2 diabetes, highlighting that further strain level information regarding type 2 diabetes progression could be useful for targeted opportunistic pathogen control or probiotic prophylaxis and treatment. This study is very interesting and well designed and performed. A good work of analysts and methods was applied, and I suggest the publication.

Our response: We are grateful to the reviewer for highlighting the strengths and  importance of  an of our manuscript and for the excellent comments below:

Comment: - Line 30: ..Lactobacillus fermentum, plantarum and casei…correct in L. plantarum and L. casei.

Our response: We apologize for the oversight. We have corrected the mistake. Please, see page 3, line 68.

Comment: - Line 76,77: …during the first study visit: healthy (n = 21; HbA1c < 5.7%), healthy but diagnosed with prediabetes (prediabetes/healthy (reversed prediabetes), n = 9; HbA1c < 5.7%)…you can improve this sentence because is not clear the difference between these two class; it has the same percentage of HbA1c.

 Our response: The HbA1c should be the same percentage, but to clarify we added the following language: “healthy range but previously diagnosed with prediabetes by a registered physician (prediabetes/healthy (reversed prediabetes), n = 9; HbA1c < 5.7%)”. Please, see page 4, lines 117-118.

Comment: - I suggest you make an image with the results found, that explain easily the difference found in different cohorts of study, within resalt the taxa significative differences.

Our response: Figure 3 constitutes the most simplified manor of summarizing the significant differences in taxa between different cohorts of the study. We have added a graphical abstract to easily explain the concept of the manuscript.

Round 2

Reviewer 1 Report

Dear authors,

Thanks to consider almost all my comments in your final version of your manuscript.

Only a higher number of cases involved in the study could have increased the power of the study.

The document can be included in the Microorganisms journal in its present form.